

# Comparison of influence between two types of cold surge
# on haze dispersion in Eastern China
Shiyue Zhang[1], Gang Zeng[1], Xiaoye Yang[1], Ruixi Wu[2], Zhicong Yin[1, 3]
[1] Key Laboratory of Meteorological Disaster of Ministry of Education (KLME),
Collaborative Innovation Center on Forecast and Evaluation of Meteorological Disasters (CIC-FEMD),
Nanjing University of Information Science and Technology, Nanjing, 210044, China
[2] Meteorological Bureau of Jiading District, Shanghai 201815, China
[3] Southern Marine Science and Engineering Guangdong Laboratory (Zhuhai), Zhuhai, 519080, China
*Correspondence to*: Gang Zeng (zenggang@nuist.edu.cn)
**Abstract.** Cold surge (CS) is considered as a favorable weather process to improve air quality and is
widely recognized. However, there is no detailed study on the differences in the dispersion ability of
different types of CSs to haze days in eastern China ($HD_{EC}$). This paper uses the hierarchical clustering
algorithm to classify the cool season (November to February of the following year) CSs across eastern
China into blocking CSs and wave-train CSs and compares their influences on the number of $HD_{EC}$ from
1980 to 2017. Results show that the wave-train CSs can significantly improve the visibility in eastern
China and generally make the high air quality last for about 2 days longer than the blocking CSs, which
indicates that the blocking CSs have a weaker ability to dissipate $HD_{EC}$ compared with the wave-train
CSs. The CSs affect the $HD_{EC}$ by changing these meteorological elements like thermal inversion potential,
horizontal surface wind, sea level pressure (SLP), and surface air temperature (SAT). 4 days after the
CSs outbreak, the variations of thermal inversion potential and horizontal surface wind of two types of
CSs tend to be consistent. However, the negative SAT anomalies, and the positive SLP anomalies caused
by the blocking CSs lasted shorter than those caused by the wave-train CSs, which forms favorable
conditions for the rapid growth of $HD_{EC}$. Furthermore, results show that in recent years, especially after
the 1990s, the frequency of wave-train CSs has decreased significantly, while the frequency of blocking
CSs has slightly increased, indicating that the overall ability of CSs to dissipate $HD_{EC}$ has weakened in
general.



## 1. Introduction


Haze can reduce visibility and affect traffic and ecological sustainability (Xu et al., 2013; Xie et al.,
2014; Wang et al., 2016). Studies have shown that the haze in China is mainly concentrated in the eastern
region of China (EC), and its peak is noticeable in winter and spring (Wang et al., 2015, 2016). During
haze days, the concentration of aerosol particles increases and results in a wide range of visibility decline
(Luo et al., 2001; Xu, 2001; Wu et al., 2012; Fu et al., 2013; Wu et al., 2014). For example, in the winter
of 2015, severe haze in the Beijing-Tianjin-Hebei region affected more than 500,000 square kilometers,
causing heavy pollution in 37 cities (Chang et al., 2016; Zhang et al., 2016). After this event, researchers
and policymakers paid more attention to the studies related to haze events. Besides, strict control
measures of air pollution and energy emissions have also been put in place.
Many studies indicated that the long-term trends of haze are closely related to fossil-fuel emissions
(Shi et al., 2008; Wei et al., 2017). On the other hand, meteorological conditions also play an important
role in determining regional air quality. In addition to the influence of human activities, the formation of
haze is closely related to static and calm weather conditions, such as strong thermal inversion potential
(TIP), negative sea level pressure (SLP) anomaly, and weak wind speed (Niu et al., 2010; Cai et al.,
2017). In recent years, due to the decreased relative humidity, it is difficult for haze particles to transform
into fog drops, making the number of haze days present a rising trend (Ding and Liu, 2014). In addition,
the anomalies of atmospheric circulation caused by global warming may also enhance the stability of the
lower atmosphere, which leads to more severe and frequent haze pollution (Cai et al., 2017). All these
emphasize that the threat of haze to human society could be more serious in the near future.
Global warming leads to the decrease of cold days and cold surges (CSs) by raising the surface air
temperature (SAT), which also provides favorable conditions for the increase of haze days (Lin et al.,
2009). CS is a typical extreme weather process in East Asia, which significantly impacts the atmospheric
circulation to improve the local air quality (Hu et al., 2000; Qu et al., 2015; Wang et al., 2016). With the
outbreak of CSs, a series of abrupt variations of meteorological elements such as the positive SLP
anomaly, the decrease of SAT, and the enhancement of north wind component will occur in the areas
where the CSs pass (CCiM et al., 1999). When a CS occurs, the arrival of fresh and dry cold air can
dissipate and reduce local air pollutants (Lin et al., 2008). Wang et al. (2016) proposed that the " early in
the north and late in the south " feature of air quality improvement in mainland China results from the





cold air masses moving southward from high latitudes to low latitudes after the outbreak of CSs.
Although some studies have shown that the weakening of East Asian Winter Monsoon and global
warming leads to the decrease of CSs (Qu et al., 2015; Wang et al., 2006), extreme low-temperature
events are still frequent (Park et al., 2011a), which makes the assessment of haze dispersion capacity of
cold air activities still full of uncertainty.
Previous studies have shown that the outbreak of CSs has an obvious effect on haze dispersion (Lin
et al., 2009; Hien et al., 2011; Ashfold et al., 2017). However, most of them analyzed the haze variation
during the CSs based on case analyses or considering the interannual influence of CS frequency on haze.
Studies have shown that there are large differences between individual cases of CSs in terms of
circulation anomalies, influence path and range (Park et al., 2014; Cai et al., 2019). Therefore, it is
necessary to consider the influence of classified CSs on haze. Based on this limitation, the following two
questions are proposed in this paper: Are there different effects of CSs' types on the haze days in EC
($HD_{EC}$)? If so, what is the physical mechanism that makes the difference? The solution to these issues
will help us understand the mechanism of CSs in dissipating the haze and improving its predictability in
the future.
The rest of this paper is organized as follows: Section 2 introduces the data and methods, while
section 3 presents the study findings. The variation of $HD_{EC}$ and its relationship with two types of CSs
are shown in section 3.1. Section 3.2 explains the reason why different types of CSs have different
abilities to dissipate $HD_{EC}$. The main conclusions and discussion are presented in section 4.
**2.  Data and Methods**
**2.1  Data**
The datasets employed in this study were: (1) daily ERA-Interim atmospheric fields including SLP,
air temperature at different levels, SAT, horizontal wind, and geopotential height (GPH) provided by the
European Center for Medium-Range Weather Forecasts (ECMWF) (Dee et al., 2011). They have a
horizontal resolution of 1.0°×1.0°. (2) daily observational datasets for 756 meteorological stations from
1980 to 2017 collected by the National Meteorological Information Center of China Meteorological
Administration, including relative humidity, visibility, and weather phenomena. These datasets were
observed four times per day (02:00, 08:00, 14:00, and 20:00LT). Stations with more than 5% missing



data were eliminated, while sporadic missing data were filled by cubic spline interpolation. Successive
missing data were discarded.
**2.2  Methods**

88        The visibility and relative humidity (Rhum) are routinely used in meteorology to distinguish the

haze (Yin et al., 2017). After filtering the other weather parameters affecting visibility (i.e., dust,
precipitation, sandstorm), we defined a haze day as a day with visibility lower than 10 km and the Rhum
less than 90 % occurring at any of the four times (02:00, 08:00, 14:00, and 20:00LT) (Yin et al., 2019a).
Figure S1 shows the climatology of haze days in China from 1980 to 2017. The haze days are mainly
concentrated in the EC (22°N-37°N, 106°E-121°E), which is selected as the target area in the present
study. The monthly average of $HD_{EC}$ indicated that the $HD_{EC}$ mainly peaks (Figure S1b) in the cool
season (November to February of the following year (NDJF)). Consequently, we chose cool season as
the study period for $HD_{EC}$ in this research.

97        The CS is a cooling process superimposed on a cold day (Park et al., 2011a). The outbreak of the

CSs in East Asia is closely related to the Siberian high, known as the Siberian high surge (Compo et al.,
1999). In this study, we first divided EC into 5°×5° grid boxes as shown in Figure 1 and then calculated
the average SAT for each box to avoid the extreme SAT anomaly in a single grid. To explore the impact
of CSs on $HD_{EC}$, the selection of CS in this paper fulfil the following three criteria (Park et al., 2011b) :
(1) the maximum pressure center in the domain of the Siberian high (Figure 1) should exceed 1,035 hPa
on the day of the CS outbreak. (2) the daily temperature drop ($SAT_t - SAT_{t-1}$) and the SAT anomalies
should exceed −1.5 standard deviation ( i.e., the standard deviation of the SAT from 1980 to 2017) at
least one box. (3) the haze day appeared in the box where the CS occurs from -2 days to 0 day related to
the occurred CS. A total of 187 CSs were identified in this paper.



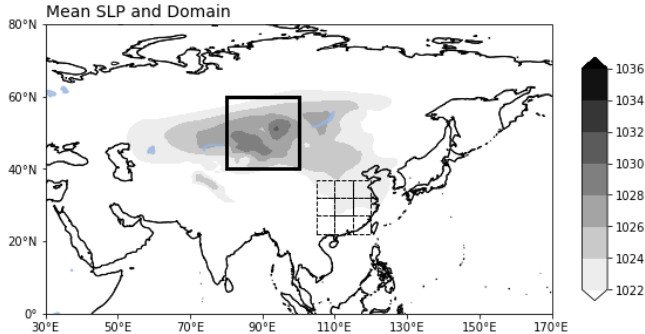


**Figure 1.** The domain of Siberian high (thick solid box; 40°N–60°N, 80°E–100°E) and EC (dotted box; 22°N-37°N,

106°E-121°E) divided into 5°× 5° grid boxes. Shadings indicate the cool season mean SLP.




**3.  Results**
**3.1 The influence of two types of CSs on $HD_{EC}$**
The circulation evolution with different types of CSs is quite different (Park et al., 2013), which
leads to the different distribution of surface meteorological conditions and haze. Here we display the
evolutions of two typical CS events (Figure 2). These two events were selected because they belong to
different types of CSs, referring to Park et al. (2008 and 2014), and have a large different effect on $HD_{EC}$.
Figures 2a, 2e, 2i, 2m show a CS that occurred on December 31, 2000, with positive and negative GPH
anomalies over the sub-arctic and East Asian coast, respectively, which meet the definition of the
blocking CS (Park et al., 2015). The blocking structure has a relatively stable lifecycle, so the $HD_{EC}$ only
has a certain dispersion on the day of the CS outbreak, and heavy $HD_{EC}$ begins to emerge 2 days after
the CS outbreaks (Figures 2j and 2n). Figures 2c, 2g, 2k, 2o indicate a CS that occurred on January 7,
1983, which meets the definition of the wave-train CS (Chai et al., 2002; Park et al., 2015). The CS is
associated with the wave-train structure of "-+ - +" at the upper troposphere. The cold air moves from
west to east and invades EC along with this zonal wave-train (Yang et al., 2020a). The wave-train CS has
a better ability to disperse $HD_{EC}$, and there is no new $HD_{EC}$ appear for a long time after the wave-train
CS erupts (Figures 2l and 2p).

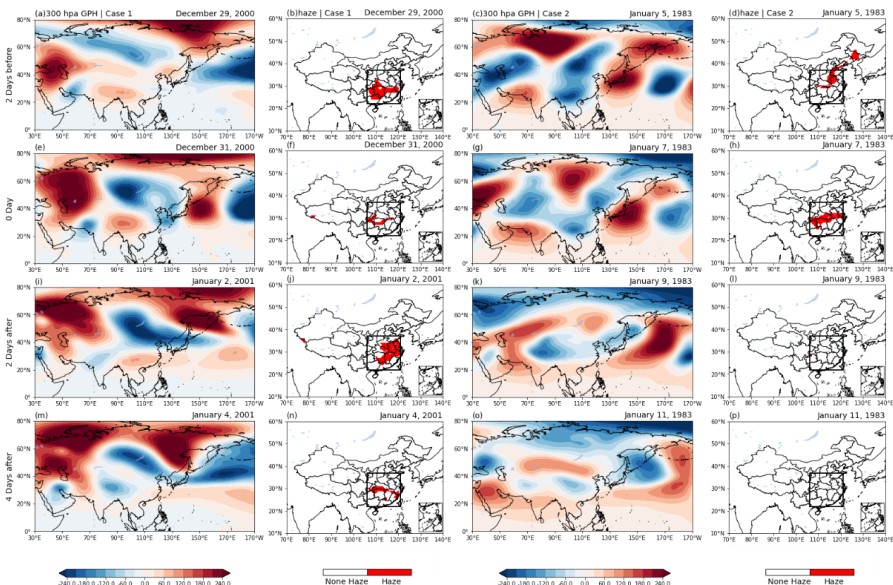


**Figure 2.** Composite of GPH anomalies (shading; gpm) at 300 hPa from -2 days to 4 days for case1 CS outbreak on

December 31, 2000 (a, e, i, m), and case2 CS outbreak on January 7, 1983 (c, g, k, o), and the related spatial

distribution of $HD_{EC}$ (shading) (b, f, j, n and d, h, l, p).

The case analysis results indicate that the ability of different types of CSs to haze dispersion is

different. Referring to the research of Park et al. (2008) and Yang et al. (2020b), this paper uses the

hierarchical clustering algorithm (HCA) to classify the CS types. The HCA (Rokach et al., 2005) creates

a hierarchical nested clustering tree by calculating the similarity between different categories of data

samples. In the clustering tree, the original data samples of different types are at the lowest level of the

tree, and the top level of the tree is the root point of a cluster. This paper uses Euclidean distance to

calculate the distance (similarity) between different samples. Here, we introduce the silhouette coefficient

to determine the best classification number (Rousseeuw, 1987). For any sample $i$, the silhouette

coefficient $s(i)$ is defined as:

$$s(i) = \frac{b(i)-a(i)}{max\{a(i),b(i)\}} \tag{1}$$

$a(i)$ means the average distance from sample $i$ to all other samples in the cluster it belongs to, and $b(i)$

means the lowest average distance from sample $i$ to all samples in any other cluster. The silhouette

coefficient of the clustering result is the average of the silhouette coefficients of all samples. The range

of silhouette coefficient is - 1 to 1. The closer to 1, the better the classification results.

According to the principle of maximum distance between clusters, the CSs from 1980-2017 can be
classified into two categories (Figure 3a). The silhouette coefficient of the clustering model shows that
when all CSs are divided into two types, the difference between them is the largest. Figures 3c and 3d
show the composite GPH anomalies at 300 hPa that depict the blocking CSs and wave-train CSs. Such
classification results are consistent with previous studies (Park et al., 2014, 2015). The cold air of the
blocking CSs mainly moves in a north-south direction that invades from Siberia to EC, and the cold air
of the wave-train CSs originating from the Ural Mountains converged near Lake Baikal then invaded EC.

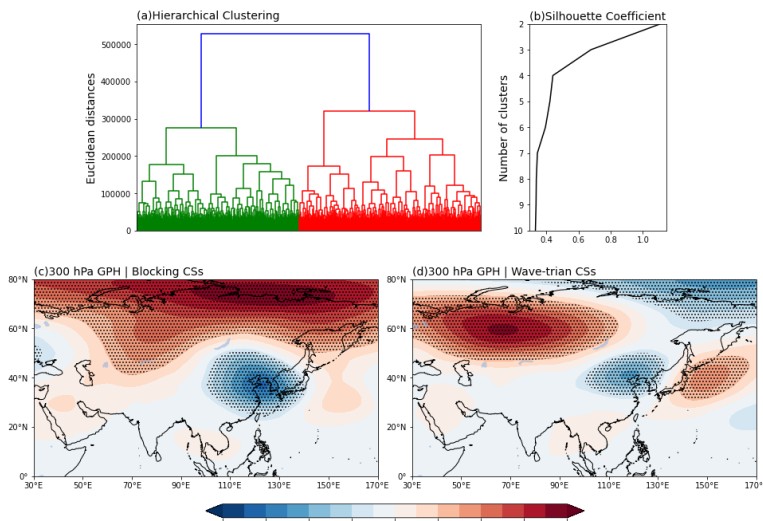


**Figure 3.** Hierarchical clustering tree (a) and silhouette coefficient (b) of cool season CSs in the EC. Composite of
GPH anomalies at 300 hPa (shading; gpm) relative to blocking CSs (c) and wave-train CSs (d). Dotted areas are
statistically significant at the 95% confidence level.

Figure 4 presents the circulation anomalies from -2 days to 6 days of the CSs events in the two types
and the related evolution of $HD_{EC}$. For the blocking CSs, largely positive and negative GPH anomalies
at 300 hPa are found over the Arctic and EC. The $HD_{EC}$ tends to dissipate first and then increase rapidly
after the CSs erupt. 6 days after the CSs erupt, the haze reaches a relatively large value (Figure 4i). For
the wave-train CSs, a zonal wave-train structure of GPH anomalies can be seen in the midlatitude of the
Eurasian. From -2 days to 6 days, the wave-train with northwest-eastern direction appears to move toward
EC. With the movement of the wave-train, the haze dissipates rapidly, and EC can maintain high air
quality weather for a longer time. Sporadic $HD_{EC}$ does not appear until 6 days after, which is different



from the existence of $HD_{EC}$ when the blocking CSs occur. It shows that blocking CSs have a weak ability
to dissipate haze compared with wave-train CSs. This conclusion is also consistent with the individual
cases mentioned above. In addition, we also used PM2.5 concentration data (acquired from the China
National Environmental Monitoring Centre and were widely used in the research of PM2.5 in China,
refer to Yin et al. (2021) and Wang et al. (2021)) together with NCEP/NCAR Reanalysis datasets (Kalnay
et al., 1996) to verify the response of PM2.5 to the two types of CSs from 2014 to 2019, and the similar
results were obtained (Figure S2). It shows that the selection of data sets does not affect the main
conclusions of this paper.

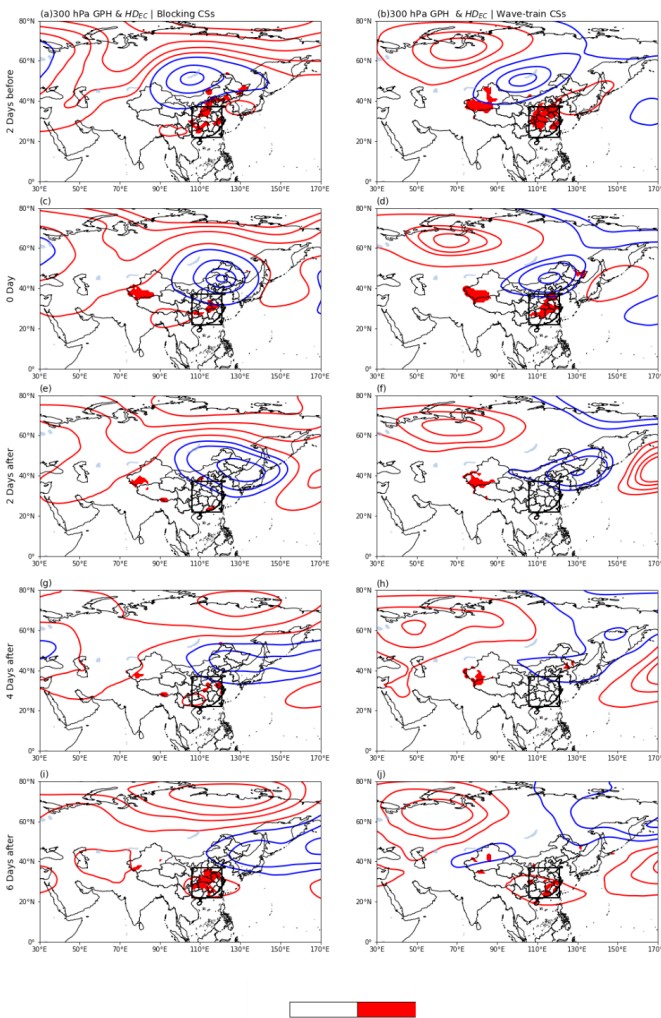


**Figure 4.** Composite of GPH anomalies at 300 hPa (contour; in intervals of 20 gpm) from -2 days to 6 days relative



to blocking CSs and the corresponding spatial distribution of $HD_{EC}$ (shading, only shows the areas which are
statistically significant at the 95% confidence level.) (a, c, e, g, i). b, d, f, h, and j are same as a, c, e, g, and i, but for
wave-train CSs.

**3.2 Why are two types of CSs different in dispersing $HD_{EC}$?**
According to the definition of $HD_{EC}$, which combines visibility and Rhum in this study, we
composite the daily visibility anomalies and Rhum anomalies for 9 days before and after the outbreak of
the blocking CSs and wave-train CSs, respectively (Figure 5). This helps to understand why the two
types of CSs have different abilities to disperse $HD_{EC}$. According to our definition, haze is determined
by visibility and Rhum. Considering two types of CSs, it was found that there is no significant difference
in Rhum between the two kinds of CSs. However, the blocking CSs are generally less effective in
improving visibility than the wave-train CSs. On the day when the blocking CSs outbreak, the visibility
shows a rising trend; however, it begins to deteriorate continuously 3 days after. Though the visibility in
EC has a noticeable downward trend 5 days before the outbreak of the wave-train CSs, it improves
significantly on the day of the wave-train CSs outbreak and rapidly deteriorates again about 3 days after
the wave-train CSs occur. Combined with the differences in the circulation evolution during the two
types of CSs shown in Figure 4, the weak dissipating ability to block CSs during $HD_{EC}$ may be related
to the stable blocking anomalies, while the significant cyclical variations of visibility during the wave-
train CSs may be related to the "+-+" wave-train structure anomalies transporting rapidly to the eastern.

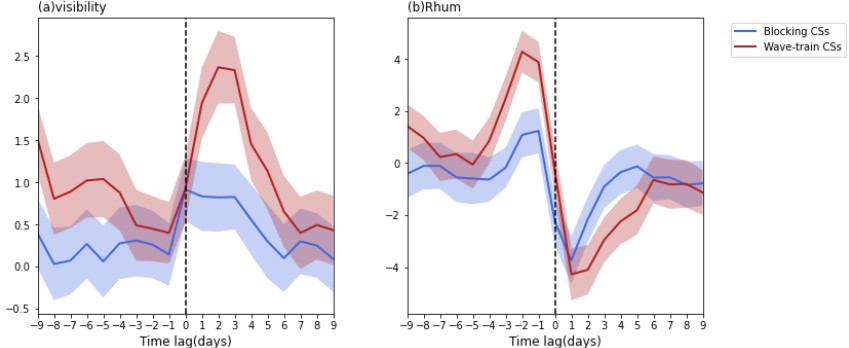


**Figure 5.** Mean (a) visibility anomalies (km), and (b) Rhum anomalies (%) in EC during 9 days before and after the
outbreak of the blocking CSs (blue lines) and wave-train CSs (red lines), respectively. Shading represents plus/minus
one standard deviation among the CSs.




Previous studies show that haze is influenced by surface meteorological conditions (Wang et al.,
2015a; Yin et al., 2019a), which have significant variations after the outbreak of CSs (Park et al., 2014).
Figure 6 reveals the thermal inversion potential anomalies (TIP, defined as the air temperature at 850 hPa
minus SAT referring to Yin et al. (2019b)), surface horizontal wind speed (UV_sfc) anomalies, SAT
anomalies, and SLP anomalies for 4 days before and after the outbreak of the blocking CSs and wave-
train CSs, respectively. The results show that the high variations of meteorological elements reached the
strongest on the day of the CSs outbreak, and their anomalies weakened in the next 4 days. Compared
with the blocking CSs, the variation of meteorological elements related to wave-train CSs is stronger. 4
days after the outbreak of the two types of CSs, the difference of TIP and UV_sfc between the two types
of CSs tended to be the same. However, the negative SAT anomalies and the positive SLP anomalies
caused by the wave-train CSs lasted longer than those caused by the blocking CSs. This is in line with
the difference in $HD_{EC}$ dispersion ability between the two types of CSs.
It should be noted that the SAT and SLP anomalies caused by the two types of CSs in this paper are
different from those of Park et al. (2014). It is due to the regions they selected to identified CSs included
the northern part of Northeast Asia. The invasion of cold air is generally from north to south, so their
research covers more CSs in Northeast Asia, while our study only focuses on CSs in eastern China with
heavy haze. If we choose the same region to identify, similar results can be obtained (Figure S3).

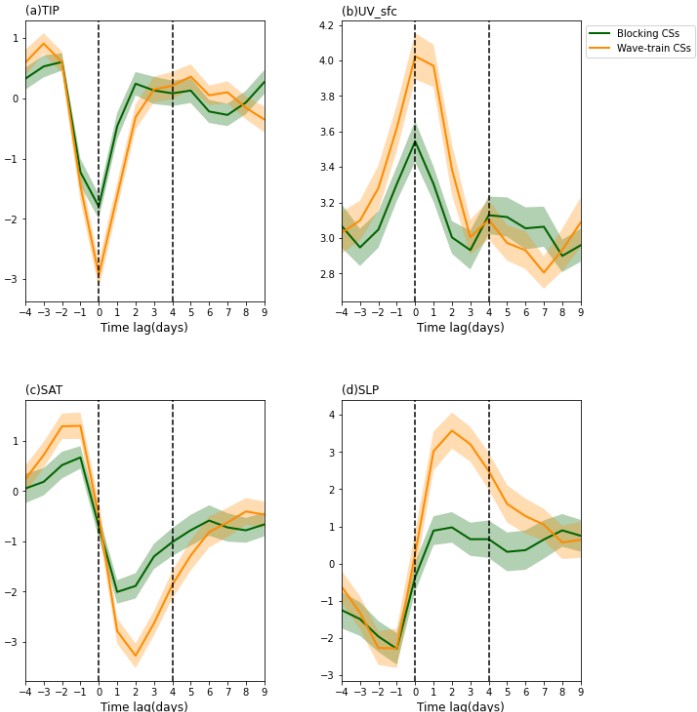

**Figure 6.** Mean (a) TIP anomalies (K), (b) UV_sfc anomalies (m s⁻¹), (c) SAT anomalies (K), and (d) SLP anomalies (hPa) in EC during 9 days before and after the outbreak of the blocking CSs (blue lines) and wave-train CSs (red lines), respectively. Shading represents plus/minus one standard deviation among the CSs.

CSs invade EC would cause a sharp drop in temperature, which may strengthen the TIP in the lower atmosphere (Lin et al., 2009). The strong TIP is unfavorable for the vertical dispersion of haze, making the cold, dry, and clear air difficult to spread (Chen et al., 2015; Zhong et al., 2019). Figure 7 indicates that the cold front will lead the TIP to control EC, forming a conducive condition to $HD_{EC}$, which may also be a reason for the rapid decline of visibility in EC 2 days after the outbreak of CSs. Compared with the wave-train CSs, the TIP after the outbreak of blocking CSs maintained for a longer time and a larger control region, which may cause the weak dispersion ability to $HD_{EC}$. From the perspective of UV_sfc, the cold air was limited to the north, and the warm and humid conditions in EC were maintained before the outbreak of the CSs. The CSs cause the airflow with the northern wind component to invade EC, rapidly dispersing the haze and causing the visibility to rise. However, after the outbreak of the CSs, the anomalies of UV_sfc in EC decrease abnormally, providing conducive conditions to the generation and





233    maintenance of haze. The anomalies of UV_sfc in EC after the outbreak of blocking CSs are weaker and

234    have a shorter duration than wave-train CSs.

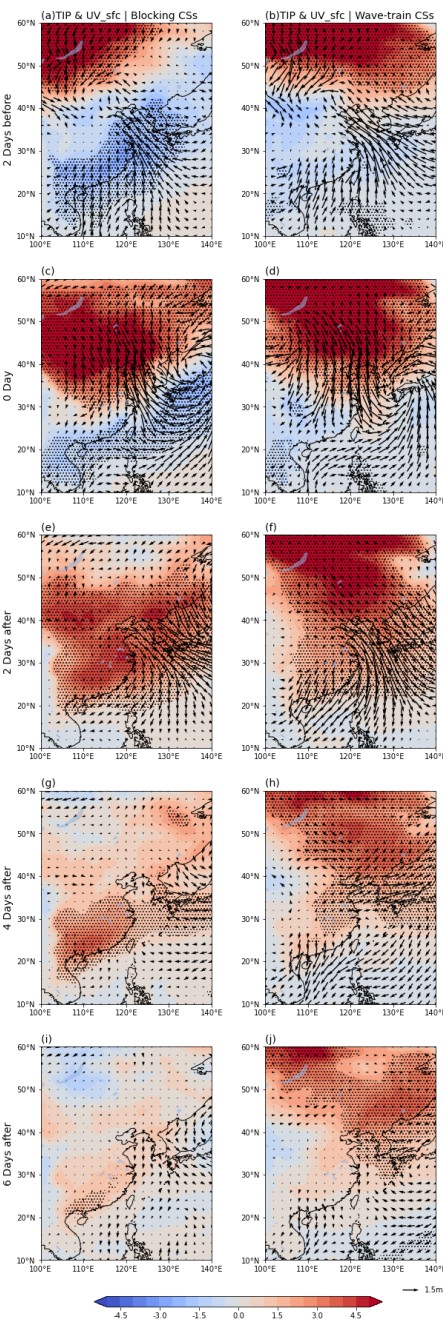

235



**Figure 7.** Composite anomalies of TIP (shading; K; dotted areas are statistically significant at the 95% confidence level) and UV_sfc (vectors; m s$^{-1}$) from -2 days to 6 days relative to the outbreak of blocking CSs (a, c, e, g, i) and the wave-train CSs (b, d, f, h, j).

The increase of Siberian high accompanies the outbreak of CSs and the splitting and southward movement of the Siberian high leads cold air into EC. Comparatively speaking, the distribution of SLP anomalies in Eurasia before the blocking CSs form a pattern similar to the negative phase of the Arctic oscillation. Figure 8 shows that when the blocking CSs occur, the positive SLP anomalies and the negative SAT anomalies in the high-latitudes move southward, and the positive SLP anomalies control EC. 2 days after the outbreak of the blocking CSs, the positive SLP anomalies, and the negative SAT anomalies in the EC decline rapidly, providing favorable conditions for the accumulation of pollutants. The occurrence of wave-train CSs is accompanied by the eastward movement of significant positive SLP anomalies and negative SAT anomalies. 2 days after, the positive SLP anomalies affect EC continuously, resulting in a longer period of high visibility in EC.

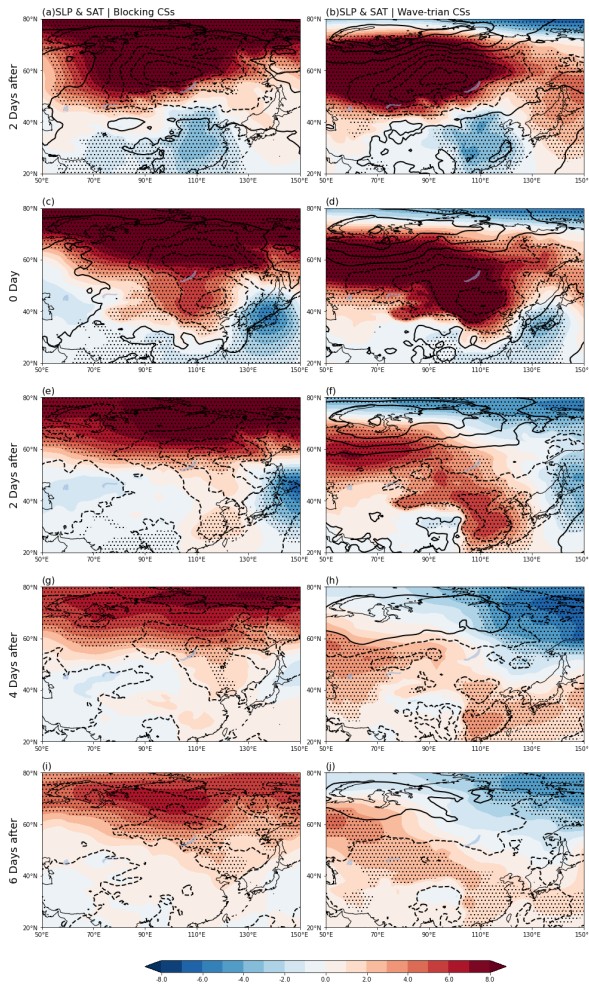

249

**Figure 8.** Composite anomalies of SLP (shading; hPa; dotted areas are statistically significant at the 95% confidence

level) and SAT (contour; K) from -2 days to 6 days relative to the blocking CSs (a, c, e, g, i) and the wave-train CSs

(b, d, f, h, j).

The results discussed earlier indicate that the blocking CSs have a weak ability to dissipate the

$HD_{EC}$, while the outbreak of wave-train CSs can make EC maintain high air quality for a longer time.

Thus, the frequency variations of the two types of CSs may also affect the trend of $HD_{EC}$ in recent years.

Figure 9a displays the time series of the frequency of blocking CSs and wave-train CSs. The results show

that the wave-train CSs have an obvious downward trend, and the blocking CSs have a slight upward

trend. To get a better sense of the variation of the two types of CSs, the ratio of blocking CSs to wave-





train CSs is shown in Figure 9b, which has a visible upward trend, and its 9-year moving average exhibits
a significant interdecadal variation. It means that in recent years, the ability of CSs to dissipate $HD_{EC}$ has
decreased in general.

263         We further calculated the partial correlation coefficients between the frequency of blocking CSs

(wave-train CSs) and $HD_{EC}$ to exclude the influence of another type of CSs. It can be found that there is
a significant positive correlation between blocking CSs and $HD_{EC}$ (Figure 9c). It should be noted that this
does not mean that more blocking CSs cause more haze but reflects the weak dispersion ability of
blocking CSs to $HD_{EC}$, resulting in relatively more $HD_{EC}$. The negative correlation between wave-train
CSs and $HD_{EC}$ is significant (Figure 9d), which is consistent with the result above. Furthermore, previous
studies have shown that with the appearance of a warm Arctic-cold Eurasian pattern, there will be more
blocking high maintained in the winter (Cohen et al., 2014; Luo et al., 2016), causing more blocking CSs
to invade East Asia, and the phenomenon is further intensified under global warming (Yang et al., 2020b).
It can be considered that ability of CSs to dissipate haze in East Asia weakened in the future, and
policymakers are required to consider the problem of air pollutions.

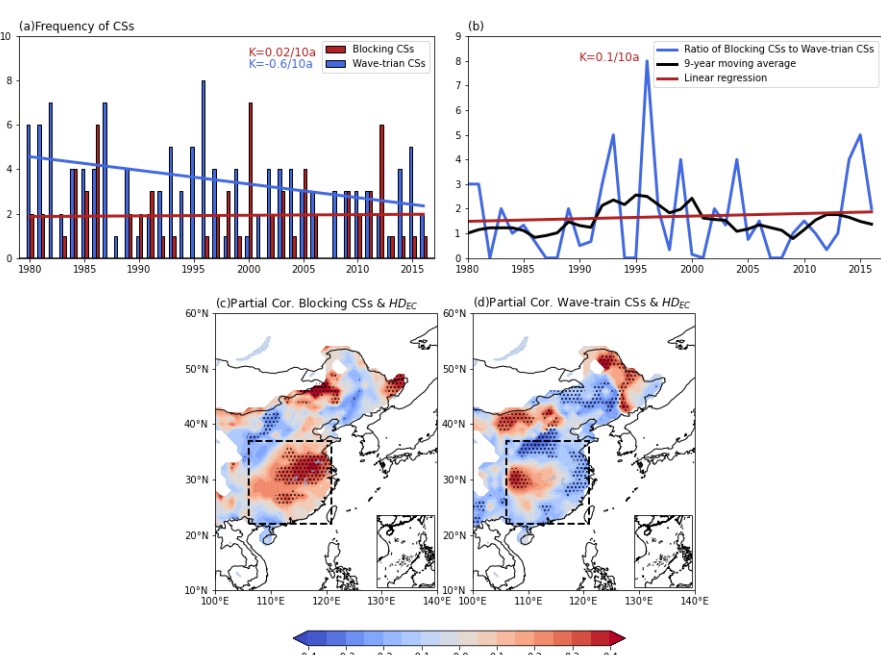


**Figure 9.** (a) Time series of the blocking CSs and the wave-train CSs (The solid line is the linear regression of the
time series, and the text in the upper right corner indicates the trend of the solid lines). (b) Time series of the ratio of


blocking CSs to wave-train CSs (blue), 9-year moving average (black), and linear regression (red). The partial
correlation coefficient between $HD_{EC}$ and the frequency of blocking CSs (c) and the wave-train CSs (d). Dotted areas
are statistically significant at the 95% confidence level.
**4.   Conclusions and discussion**
In this paper, the connection between the CSs and the cool season haze over the EC is investigated
based on the observational and reanalysis datasets from 1980 to 2017. The 187 CSs over EC are classified
into two types by HCA, blocking CSs and wave-train CSs. Usually, the blocking CSs are accompanied
by a meridional dipole in the upper-tropospheric GPH anomalies, which consists of a stable blocking
structure. The blocking structure tends to control the EC for a long time and forms a relatively stable
meteorological condition, which has the disadvantage to dissipate the $HD_{EC}$. Correspondingly, the local
meteorological conditions, especially TIP and the quiescent wind band, rapidly appear after the blocking
CSs outbreak, provide a haze-prone background. In addition, the positive SLP anomalies induced by the
outbreak of the blocking CSs can rapidly restore to normal, and the SAT warm up under the influence of
the weakening of the north wind component. Therefore, the ability to block CSs to dissipate $HD_{EC}$ is
limited. On the contrary, high air quality in EC can last longer due to the shorter duration of TIP and
longer duration of positive SLP anomalies. In general, $HD_{EC}$ can remain at a low level for a shorter
(longer) time after the outbreak of blocking (wave-train) CSs. It is confirmed that blocking CSs has been
increased over the past few years (Park et al., 2011a; Luo et al., 2018). Furthermore, the increasing trend
of blocking CSs is likely to continue in the future, which may weaken the dispersion of haze and worsen
the $HD_{EC}$. This reminds us that the problem of air pollution is still very serious and needs more attention.
Finally, it should be noted that the lack of meteorological station information in the observed data
limits the accuracy of the haze distribution to some extent. Previous studies documented that the CSs
outbreak greatly affected the dispersion of haze (Lin et al., 2009 and Wang et al., 2016). However, it can
be seen that the visibility anomalies from -5 days to the day of the wave-train CSs outbreak have a sharp
decline trend. Whether the CSs aggravate the haze before the outbreak deserves our attention in future
research. In addition, to better understand the formation of haze and improve the predictability of haze,
more research is needed to explore the possible impacts of meteorological elements on haze pollution in
China.




**Data availability**

The ground observations are from the website: http://data.cma.cn. Daily mean meteorological data
are obtained from the ERA-Interim reanalysis data archive: http://www.ecmwf.int/en/research/climate-
reanalysis/era-interim.

**Author contributions**

SZ and GZ put forward the idea and design research, RW provided observational data including
relative humidity, visibility, and weather phenomena. SZ and XY performed research, and ZY provided
valuable suggestions. SZ wrote the manuscript with contributions from all co-authors.

**Competing interests**

The authors declare that they have no conflict of interest.

**Acknowledgements**

This research is supported by the National Key Research and Development Program of China
(2017YFA0603804) and the National Natural Science Foundation of China (41575085).

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
