# Peer review of "Comparison of influence between two types of cold surge"

_Atmospheric Chemistry and Physics, 2021_

## Author Comment (AC1)

**Response to the Comments**

Dear reviewer,

We thank you so much for taking time to enhance the quality of our paper. We have revised the manuscript, and changes are shown with red color in the revised manuscript. Below are our responses to the reviewers' comments. All reviewers' comments are in black, while the authors' responses are in blue. And all revisions in the revised manuscript are highlighted in red color.

**General comments**

This manuscript addresses interesting questions that are within the scope of ACP - how do different types of cold surge contribute to the dissipation of haze in Eastern China during the northern hemisphere winter, and are there trends in these contributions? The findings contain information on varying trends in different types of cold surge which has implications for understanding how air pollution levels in this region will change in future.

On the whole the methodology, the results and the interpretation appear largely sound. However, the methods need to be more fully and carefully explained, including more information on how this work builds upon the methodological approaches of past research. I also make suggestions below that should help to structure material within the manuscript and to clarify various details of the results and interpretation.

The fluency and precision of the language used could be improved and would certainly benefit from further work. Included in the specific and technical comments below are suggestions for changes which would help to clarify meaning.

Also throughout the Figure labelling is with a font size that is smaller than ideal - it would be helpful to increase the font size significantly for a later version of this manuscript. Figures 2, 4, 7, 8 in the current layout were especially hard to read.

**Specific comments**

(1) Line 26 – add a sentence summarising new implications from this work; any indication of trend being due to changing climate, or how the identified trend contributes to an overall literature-based view on haze events in future; given past work, is this work in agreement overall?

**Response:** Thank you for the suggestion. The following statement was added in the revised manuscript: This work may provide reference for the future formulation of haze control policies in East Asia. Please refer to lines 25-26 in the revised manuscript.

This research discussed the trend of the two types of cold surges in the past 37 years. The results show that the ability of blocking cold surges to dissipate haze is generally weaker than that of wave-train cold surges. The previous study has proposed that wave-train cold surges will show a significant downward trend in the 21st century, and the frequency of blocking cold surges will remain stable (Yang et al., 2020), indicating that the number of wave-train cold surges which can significantly dissipate haze will greatly reduce in the 21st century. In general, the ability of East Asian cold surges to dissipate haze is declining. Therefore, we have made the following revision in the revised manuscript (see lines 299-304):

Furthermore, previous studies suggested that with the appearance of a warm Arctic-cold Eurasian pattern, more blocking high are expected to be maintained in the winter (Cohen et al., 2014; Luo et al., 2016), and the significant downward trends in wave-train cold surges are projected in the 21st century (Yang et al., 2020b), indicating that the number of wave-train cold surges which can significantly dissipate haze is likely to reduce in the 21st century. Thus, in general, the ability of East Asian cold surges to dissipate haze is declining, and policymakers are required to consider the problem of air pollutions.

This study is consistent with previous studies (Park et al., 2015; Yang et al., 2020), especially the circulation evolution and the variations of frequency of the two types of cold surges.

**References:**

Park, T. W, Ho, C. H, Jeong, J. H, et al.: A new dynamical index for classification of cold surge types over East Asia, Climate dynamics, 45(9): 2469-2484, https://doi.org/10.1007/s00382-015-2483-7, 2015.

Yang, X. Y., Zeng, G., Zhang, G. W., Vedaste, I., Xu, Y.: Future projections of winter cold surge paths over East Asia from CMIP6 models, International Journal of Climatology, 1-16, https://doi.org/10.1002/joc.6797, 2020.

(2) 29 – it would be useful to explain briefly that "haze" is used in referring to a specific type of pollution event in China. The term "haze" can refer to slightly different types of pollution in different regions.

**Response:** According to your suggestion, we have made the following modifications to the corresponding sentences in the revised manuscript (see lines 29-31): Haze in eastern China is generally referred to as the polluted particulate aerosols suspended in the air (Yin et al., 2019a), and can reduce visibility and affect traffic and ecological sustainability (Xu et al., 2013; Xie et al., 2014; Wang et al., 2016).

Considering the definition of haze in eastern China and the limitations of data, we do not explain the detailed chemical composition of haze, which is consistent with the method of previous research on haze in China.

(3) 67 – "it is necessary to consider the influence of classified CSs on haze" – this statement can be elaborated on, especially whether classification has been used in previous studies on cold surges and haze, and if so, how are the CS types referred to on line 68 (and 73) linked to the literature?

**Response:** Thank you for the suggestion. Few studies discussed the impact of different types of cold surges on haze in China, which is the main topic of this paper. However, previous studies have clearly discussed the classification of cold surges in the study area (based on the circulation anomalies, the cold surges are divided into wave-train cold surges and blocking cold surges). By comparing circulation anomalies together with the cold surge index designed by Park et al (2015), previous studies (Park et al., 2014, Yang et al., 2020) used the same classification of CSs. Therefore, the present study adopts the same category, and the results are consistent with previous work. Thus, to elaborate more on the classification of CSs, description is given in section 2.4, and the following statement is added in the revised manuscript (see lines 72-73): The variation of $HD_{EC}$ and its relationship with different types of CSs are described in section 3.1.

**References:**

Park, T. W, Ho, C. H, Jeong, J. H, et al.: A new dynamical index for classification of cold surge types over East Asia, Climate dynamics, 45(9): 2469-2484, https://doi.org/10.1007/s00382-015-2483-7, 2015.

Park, T. W., Ho, C. H., Deng, Y.: A synoptic and dynamical characterization of wave-train and blocking cold surge over East Asia, Climate dynamics 43, 753–770, https://doi.org/10.1007/s00382-013-1817-6, 2014.

Yang, X. Y., Zeng, G., Zhang, G. W., Li, Z.X.: Interdecadal Variation of Winter Cold Surge Path in East Asia and Its Relationship with Arctic Sea Ice, Journal of Climate, 33(11), 4907–4925, https://doi.org/10.1175/JCLI-D-19-0751.1, 2020.

(4) 89 – it will be helpful to explain more fully how the other parameters (dust, sandstorm etc.) are defined and then used to filter out some days

**Response:** In addition to the visibility and relative humidity, the observation data set we use also includes weather phenomena (i.e., dust, precipitation, sandstorm), so we have eliminated the date of the weather event before calculating the haze day. The specific observing standards for these weather phenomena are stipulated by the operating specifications of observing stations issued by the China Meteorological Administration.

(5) 94 – in Figure S1b clarify how are the number of haze days in a month calculated when presumably not all locations in EC meet the criteria on the same day?

**Response:** The number of monthly average haze days is obtained from the regional average haze days in eastern China (i.e., total haze days in all grids /number of all grids). This method is generally accepted in climate research. We have made the following addition to the corresponding sentences in the revised manuscript (see lines 107-109): The number of the monthly average of $HD_{EC}$ by regional average indicated that the $HD_{EC}$ mainly peaks (Figure S1b) in the cool season (November to February of the following year (NDJF)).

(6) 106 – good to clarify the identified CS are those that might affect a haze day, i.e. "a total of 187 CSs that might affect a haze day in EC were identified"

**Response:** Thank you for your suggestion. The sentence is revised as follows (see lines 123): A total of 187 CSs that might affect a haze day in EC were identified.

(7) 133-145 – I suggest this text on CS type classification will be more appropriately moved to the preceding section on Data and Methods. It would also be helpful if the procedure could be explained a more fully, for example more clearly on how the approach and the definition of the two CS types follows from previous work, and including an explanation which data is used in the classification.

**Response:** Accepted. We move the introduction of the clustering algorithm to Section 2.4, and the following two sentences were added in the revised manuscript. Lines 140-142: In this paper, the GPH anomalies in the region of 30°E-170°E, 0°-80°N at 300 hPa on the outbreak day of CS was used to perform HCA. and see lines 170-171: Such classification results are consistent with previous studies (Park et al., 2014, 2015), which are mainly manifested in the location of the center of circulation anomalies.

(8) 150-152 – to support this text it would be helpful to indicate the movement of air (e.g. lower level wind vectors) within Figure 3

**Response:** Figures 3c and 3d are the results of classification. Since we only use GPH anomalies for clustering, we do not show the lower-level wind vectors. Secondly, the result is only the circulation anomalies on the outbreak day of the cold surge, and the evolution of the cold air movement cannot be seen. However, it has been shown in Figure 4 (the movement of the negative anomalies represents the moving direction of the cold air). In addition, the evolution of the low-level wind vector field is shown in Figure 7. Considering the difference in cold air movement shown in the clustering results, we overlay the original wind vector field of 850 hPa in Figures 3c and 3d.

[Figure]

**Figure 3.** Hierarchical clustering tree (a) and silhouette coefficient (b) of cool season CSs in the EC. Composite of GPH anomalies at 300 hPa (shading; gpm. Dotted areas are statistically significant at the 95% confidence level.) and horizontal wind anomalies at 850 hPa (vectors; m s$^{-1}$. Only shows the areas which are statistically significant at the 95% confidence level) relative to blocking CSs (c) and wave-train CSs (d).

(9) 161 – here it would be good to add a quantification of "relatively large value". Perhaps the fraction of locations categorised as haze within the EC box? This fraction could be displayed for each panel in Figure 4.

**Response:** Accepted. We add a percentage in the lower right corner of each subgraph to represent the percentage of haze area in the study area.

[Figure]

**Figure 4.** Composite of GPH anomalies at 300 hPa (contour; in intervals of 20 gpm) from day -2 to day 6 relative to the outbreak of CSs and the corresponding spatial distribution of $HD_{EC}$ (shading, only shows the areas which are statistically significant at the 95% confidence level by t-test.) for blocking CSs (a, c, e, g, i) and wave-train CSs (b, d, f, h, j). The number in the lower right corner of each figure represents the ratio of the grids of $HD_{EC}$ to that of EC.

(10) 168-173 – for this passage I suggest moving the information on the datasets to Section 2, and explaining there more fully why you use the visibility for the main analysis rather than PM2.5 – presumably the length of dataset? Within this explanation it would also be appropriate to discuss briefly how these two types of pollution indicator (visibility and PM2.5) have been used and compared in previous studies on this problem.

Here you can then just mention the consistent result of this supplemental analysis. And within Figure S2, please better explain in the caption the PM2.5 data values – presumably this is a difference in concentration from some reference quantity? The range seems small though (+/- 0.2 µg/m3)?

**Response:** Your suggestion effectively improves the logic of this article. We move the corresponding sentences to Section 2.1 and briefly explain the purpose (see lines 91-96): In addition, we also used PM2.5 concentration data (acquired from the China National Environmental Monitoring Centre and were widely used in the research of PM2.5 in China, refer to Yin et al. (2021) and Wang et al. (2021)) together with NCEP/NCAR Reanalysis datasets (Kalnay et al., 1996) to verify the response of PM2.5 to the two types of CSs from 2014 to 2019. Such a scheme can effectively avoid the dependence of conclusions on datasets and ensure that our results are based on many cold surge samples.

A study uses visibility (relative humidity) and PM2.5 (Yin et al., 2017), and shows that there is no contradiction between them. However, due to the limitation of the time range of the two datasets, few studies discussed the differences between them in detail. Only from the results of this paper, the response of visibility and PM2.5 to cold surge is consistent. Therefore, we modified the description of figure S2 as follows (see lines 190-192): In addition, we verify the response of PM2.5 to the two types of CSs from 2014 to 2019, and similar results were obtained (Figure S2). It shows that the selection of data sets does not affect the main conclusions of this paper.

Thank you for your careful inspection. We showed the composited standardized PM2.5 anomalies in Figure S2. In fact, it should not be standardized here. Therefore, we corrected the picture (The standardization of data does not change the distribution of data, so the conclusions are consistent).

[Figure]

Figure S2. Composite of GPH anomalies (shading; gpm) at 300 hPa from day -2 to day 6 for blocking CSs outbreak (a, e, i, m, q), and wave-train CSs outbreak (c, g, k, o, s), and the related spatial distribution of PM2.5 anomalies (shading; μg m-3) (b, f, j, n, r and d, h, l, p, t) from 2014 to 2019.

**Reference:**

Yin, Z. C., Wang, H. J.: Role of atmospheric circulations in haze pollution in December 2016., 17, 11673-11681, https://doi.org/10.5194/acp-17-11673-2017, 2017.

(11) 177 – please explain the test of significance to indicate haze here.

**Response:** The t-test of independent samples was used to test the significance difference between two Bernoulli distributions. We added the corresponding figure caption (see lines 194-197): Figure 4. Composite of GPH anomalies at 300 hPa (contour; in intervals of 20 gpm) from day -2 to day 6 relative to the outbreak of CSs and the corresponding spatial distribution of $HD_{EC}$ (shading, only shows the areas which are statistically significant at the 95% confidence level by t-test.) for blocking CSs (a, c, e, g, i) and wave-train CSs (b, d, f, h, j). The number in the lower right corner of each figure represents the ratio of the grids of $HD_{EC}$ to that of EC.

(12) 185 – the variations in Rhum do look similar, but it will be helpful to explain how was the difference was tested?

**Response:** The trend of Rhum during the two types of cold surge is consistent. We use one standard deviation to show the difference between the samples of the cold surge in the same type. After the outbreak of cold surge, compared with the variations of visibility, the difference of relative humidity is weak (difference between the lower limit of blocking cold surge and the upper limit of wave-train cold surge). Therefore, we supplemented the corresponding sentences (see lines 203-205): Considering two types of CSs, it was found that there is no significant difference in Rhum between the two kinds of CSs, which were reflected in the trend and difference between the lower limit of blocking CS and the upper limit of wave-train CS after the outbreak of CS.

(13) 191-194 – finishing with "… transporting pollution to the east" would be clearer. However, I think overall these lines do not add much new information. Suggest rephrasing to link to next paragraph: a closer investigation of additional

meteorological parameters is required for further understanding of the differences between the two types of CS.

**Response:** Accepted. We deleted redundant sentences and modified the cohesion between the two paragraphs (see lines 210-212): To further understand the differences between the two types of CS, a closer investigation of additional meteorological parameters was performed.

(14) 225 – in the passage "the cold front will lead the TIP to control EC" it will be helpful to explain how the cold front is identified (the edge of the positive anomaly?) and within this paragraph, "TIP" is just the parameter, would "large positive TIP values" be more precise?

**Response:** Accepted. The corresponding sentence has been revised (see lines 245-248): Figure 7 indicates that the cold front (the edge of the positive anomaly) will lead large positive TIP values to control EC, forming a conducive condition to $HD_{EC}$, which may also be a reason for the rapid decline of visibility in EC 2 days after the outbreak of CSs.

(15) 251 – with the current presentation the contours in Figure 8 indicating SAT anomalies need rather close scrutiny to interpret, so I would encourage authors to consider a clearer way of presenting these, and certainly include some indication of the size of the anomalies is needed.

**Response:** To show the distribution of temperature anomalies more clearly, we added a thick black isoline to represent the value of 0. In addition, we add a numerical label to the temperature contours to show the size of the temperature anomalies.

[Figure]

Figure 8. Composite anomalies of SLP (shading; hPa; dotted areas are statistically significant at the 95% confidence level) and SAT (contour; K) from day -2 to day 6 relative to the outbreaks of blocking CSs (a, c, e, g, i) and the wave-train CSs (b, d, f, h, j). The thick black isoline represents the 0 value of SAT anomalies.

(16) 260 – for the ratio shown in Figure 9b, is there an issue with this calculation when there are zero occurrences of one type of CS in a particular year? For example, in 1982, there are 7 wave train CS and 0 blocking CS, which is plotted as a ratio of 0 in 9b – will this not have the opposite effect on your moving average than it should? Overall I am not sure you need this ratio analysis to make the final statement on line 261-262, which already follows from 9a. However, to support this statement further it would be interesting to assess the trend in total number of cold surges (of both types) which so far is not discussed explicitly.

I also think the descriptions of Figure 9b are given the wrong way around – should it not be the ratio of wave-train to blocking CS? (for example, in 1980, the ratio plotted in 9b is 3 wave train to 1 blocking)

**Response:** We have reconsidered the calculation of the ratio, and your concern is correct. When a certain kind of cold surge does not occur in a certain year, this will cause the results to interfere with the mean state. Therefore, we deleted figure 9b and the corresponding descriptions and only use the trend of two types of cold surges to evaluate the possible impact on haze dispersion. We have revised lines 281-304 and figure 9, including the revisions of the next two questions.

The results discussed earlier indicate that the blocking CSs have a weak ability to dissipate the $HD_{EC}$, while the outbreak of wave-train CSs can make EC maintain high air quality for a longer time. Thus, the frequency variations of the two types of CSs may also affect the trend of $HD_{EC}$ in recent years. Figure 9a displays the time series of the frequency of blocking CSs and wave-train CSs. The results show that the wave-train CSs have an obvious downward trend, and the blocking CSs have a slight upward trend. It means that in recent years, the ability of CSs to dissipate $HD_{EC}$ has decreased in general. We further calculated the partial correlation coefficients between the frequency of the two types of CS and $HD_{EC}$ to exclude the influence of the other type of CSs. It is found that there is a significant positive correlation between blocking CSs and $HD_{EC}$ (Figure 9b). It should be noted that this does not mean that more blocking CSs cause

more haze but reflects the weak dispersion ability of blocking CSs to $HD_{EC}$, resulting in relatively more $HD_{EC}$. The negative correlation between wave-train CSs and $HD_{EC}$ is significant (Figure 9c), which is consistent with the result above.

In addition, we also evaluated the relationship between the trend of the total number of CSs (of both types) and $HD_{EC}$ (Figure 9d). The results show that the correlation is weaker than that between a single type of CSs and $HD_{EC}$, which is the interference caused by the difference in the ability of the two types of CSs to dissipate haze. In fact, the relatively more $HD_{EC}$ in the central EC caused by variations of total CSs could be supported in a previous study (Yang et al., 2020a). The pattern of total CSs changes from wave-train type to blocking type, especially after the mid-1990s.

Furthermore, previous studies also have shown that with the appearance of a warm Arctic-cold Eurasian pattern more blocking high is expected to be maintained in the winter (Cohen et al., 2014; Luo et al., 2016), causing the pattern of CS changed from wave-train type to blocking type (Yang et al., 2020b). Therefore, it can be considered that the ability of CSs to dissipate haze in East Asia weakened in the future is mainly due to the significant reduction of wave-train CS, and policymakers are required to consider the problem of air pollutions.

[Figure]

**Figure 9.** (a) Time series of the blocking CSs and the wave-train CSs (The solid line is the linear regression of the time series, and the text in the upper right corner indicates the trend of the solid lines). The partial correlation coefficient between $HD_{EC}$ and the frequency of blocking CSs (b) and the wave-train CSs (c), and the correlation coefficient between $HD_{EC}$ and the frequency of all CSs (d). Dotted areas are statistically significant at the 95% confidence level.

(17) 263-264 – the explanation of the partial correlation coefficient procedure could be clearer to understand exactly what was done. Would "of the two types of CS" rather than "of blocking CSs (wave-train CSs)" and "the other" rather than "another" describe the procedure correctly?

**Response:** Accepted and corrected. We have made a careful revision to fix any other language issues in the manuscript.

(18) 272 – rather than an increase in blocking CS, I think your results show the major reason for the ability of CS to dissipate haze in future would be the reduction in wave-train CS. It would be important to assess existing literature that relates to this finding that you can include in the discussion here.

**Response:** We accepted your suggestion. See the previous question (16) for details: Furthermore, previous studies also have shown that with the appearance of a warm Arctic-cold Eurasian pattern more blocking high is expected to be maintained in the winter (Cohen et al., 2014; Luo et al., 2016), causing the pattern of CS changed from wave-train type to blocking type (Yang et al., 2020b). Therefore, it can be considered that the ability of CSs to dissipate haze in East Asia weakened in the future is mainly due to the significant reduction of wave-train CS, and policymakers are required to consider the problem of air pollutions. (Line 299-304)

(19) 281 – given the proceeding Results section already contains some reasonable discussion of the results, I suggest moving the few discussion points currently in this Section 4 to the relevant part of Section 3. The label for section 3 could then be "Results and Discussion", and the label for this Section 4 could be "Conclusions".

**Response:** Accepted. We moved the few discussion points currently in this Section 4 to the relevant part of Section 3. And revised the labels of section 3 and section 4.

(20) 296 – given your results, I think in this sentence on "the future" there should be more emphasis on the strong trend for a reduction in wave-train CS

**Response:** You are right. We have revised this sentence (see lines 326-328): Furthermore, the decreasing trend of wave-train CSs is likely to continue in the future,

while the frequency of blocking CSs is expected to remain stable, which may weaken the dispersion of haze and worsen the $HD_{EC}$.

(21) 298-305 – linking to the comment on line 281, I suggest integrating this text with your discussion of relevant results. In doing so it will be helpful to expand on the limitations and draw upon insights from the literature. For example, "the lack of meteorological station information" is too vague.

**Response:** Accepted. We revised the structure of sections 3 and 4. The details are linked to the response of the comment on 'Specific comments (19)'.

**Technical corrections**

(1) Line 1 – "Comparison of the influence of two types of cold surge on haze dispersion in Eastern China" would be a better wording for the title

**Response:** Accepted.

(2) 16 – "improve air quality" rather than "make the high air quality last"

**Response:** Accepted.

(3) 18 – delete "these"

**Response:** Accepted.

(4) 54 – there is a problem with the author name formatting in the reference

**Response:** It has been revised.

(5) 83 – define precisely "weather phenomena"

**Response:** We give a more detailed explanation (see lines 83-87): daily observational datasets for 756 meteorological stations from 1980 to 2017 collected by the National Meteorological Information Center of China Meteorological Administration, including relative humidity, visibility, and weather phenomena (It refers to the physical phenomena of precipitation, surface condensation, visual range obstacle, atmospheric optics, lightning, and wind in the atmosphere and near surface).

(6) 85-86 – define precisely "sporadic" and "successive"

**Response:** We give a more detailed explanation (see lines 88-90): Stations with more than 5% missing data were eliminated, while sporadic missing data (less than 3 days) were filled by cubic spline interpolation. Successive (3 days and more) missing data were discarded.

(7) 88 – define "Rhum" on line 83

**Response:** Accepted.

(8) 125 – for clarity I suggest rephrasing as "and no $HD_{EC}$ appear for a long time" and replacing "long time" with a more quantitative typical time period (something like "several weeks")

**Response:** Accepted. We have revised the sentence (see lines 157-159): Therefore, the wave-train CS has a better ability to disperse $HD_{EC}$, and no $HD_{EC}$ appears for about a week after the wave-train CS erupts (Figures 2l and 2p).

(9) 155 – on the terminology here, would the anomalies not be relative to an average, and so "GPH anomalies at 300 hPa (shading; gpm) for blocking CSs (c) and wave-train CSs (d)" would be more accurate?

**Response:** All anomalies in this paper are relative to an average. Therefore, the comparison of GPH anomalies is unified in this paper.

(10) 161 – for great clarity write "By 6 days …"

**Response:** Accepted.

(11) 163 – "Eurasian landmass" rather than "Eurasia"

**Response:** Accepted.

(12) 163 – it was not too clear what "with northwest-eastern direction" means. Is this to do with the orientation relative to the EC box?

**Response:** Our description was ambiguous and has been corrected (see line 185): From day -2 to day 6, the zonal wave-train appears to move toward EC.

(13) 168 – for the "cases mentioned above" include a reference to Figure 2

**Response:** Accepted.

(14) 204 – does the Figure not show 4 days before and 9 days afterward?

**Response:** After about 6 days, the changes of various meteorological elements tend to be consistent. And the difference between the samples of the same type of cold surge 4 days before is huge and has no obvious comparative significance. To maintain the unity of the figures and avoid readers' misunderstanding, we revised figure 6 to unify the time range of the x-axis.

[Figure]

**Figure 6.** Regional averaged (a) TIP anomalies (K), (b) UV_sfc anomalies (m s$^{-1}$), (c) SAT anomalies (K), and (d) SLP anomalies (hPa) over EC during 9 days before and after the outbreak of the blocking CSs (blue lines) and wave-train CSs (red lines), respectively. Shading represents plus/minus one standard deviation among the CSs.

(15) 205 – delete "respectively", I don't think it applies here

**Response:** Accepted.

(16) 206-207 – suggest rephrasing this sentence for clarity: "The variation of meteorological elements during wave-train CSs is larger than during blocking CSs"

**Response:** Accepted.

(17) 210 – explain more fully why "This is in line …" by referring briefly to main mechanism and/or literature

**Response:** We give a more detailed explanation (see lines 228-230): Namely, the negative anomaly of temperature and the positive anomaly of pressure change are not conducive to the maintenance of haze (Yin et al., 2019a).

(18) 213 – for greater clarity suggest rewriting as: " … Park et al. (2014) who identified CSs in a different region which included the northern part of Northeast Asia."

**Response:** Accepted.

(19) 219 – the caption does not accurately reflect the timescale or the colour scheme in this Figure. For consistency and clarity, it would be better to keep the same colour scheme as in Figure 5.

**Response:** Accepted. We use the same colour scheme and the timescale.

[Figure]

**Figure 6.** Regional averaged (a) TIP anomalies (K), (b) UV_sfc anomalies (m s$^{-1}$), (c) SAT anomalies (K), and (d) SLP anomalies (hPa) over EC during 9 days before and after the outbreak of the blocking CSs (blue lines) and wave-train CSs (red lines), respectively. Shading represents plus/minus one standard deviation among the CSs.

(20) 222 – "invading" rather than "invades"

**Response:** Accepted.

(21) 228 – "control region" in EC, or within the plotted domain? Please clarify.

**Response:** We give a more detailed explanation (see lines 248-250): Compared with the wave-train CSs, the TIP after the outbreak of blocking CSs maintained for a longer time and a larger control region in EC, which may cause the weak dispersion ability to $HD_{EC}$.

(22) 232 – not sure if "abnormally" is the correct word here, just delete?

**Response:** Accepted.

(23) 236 – Given it is referred to in the text it would be useful in Figure 7 to mark the EC area. Same comment for Figure 8.

**Response:** We marked the EC region in Figures 7 and 8.

[Figure]

**Figure 7.** Composite anomalies of TIP (shading; K; dotted areas are statistically significant at the 95% confidence level) and UV_sfc (vectors; m s$^{-1}$) from day -2 to day 6 relative to the outbreak of blocking CSs (a, c, e, g, i) and the wave-train CSs (b, d, f, h, j).

[Figure]

**Figure 8.** Composite anomalies of SLP (shading; hPa; dotted areas are statistically significant at the 95% confidence level) and SAT (contour; K) from day -2 to day 6 relative to the outbreaks of blocking CSs (a, c, e, g, i) and the wave-train CSs (b, d, f, h, j). The thick black isoline represents the 0 value of SAT anomalies.

(24) 267 – explain relative to what

**Response:** We added this sentence (see lines 289-292): It should be noted that this does not mean that more blocking CSs cause more haze but reflects the weak dispersion ability of blocking CSs to $HD_{EC}$, resulting in relatively more $HD_{EC}$ to that caused by wave-train CSs.

(25) 268 – suggest start a new paragraph for the sentence beginning on this line

**Response:** Accepted.

(26) 276 – why are there inset maps in the lower right of Figures 9c and d? Are they needed?

**Response:** This is necessary. The main areas we highlight include the east of China, so we must give a complete map of China in the figures, which is stipulated by relevant policies.

(27) 277 – clarify in the caption that in the text on the trend 10a means, I assume, per decade?

**Response:** We change '10a' in Figure 9a to decade.

(28) 291 – "of blocking" rather than "to block"

**Response:** Accepted.

(29) 292 – is this referring to when there is a wave-train CS?

**Response:** Yes, we added the corresponding sentence (see lines 322-323): On the contrary, high air quality in EC can last longer due to the shorter duration of TIP and longer duration of positive SLP anomalies after the wave-train CS.

---

## Author Comment (AC2)

**Response to the Comments**

**Dear reviewer,**

We thank you so much for taking time to enhance the quality of our paper. We have revised the manuscript, and changes are shown with red color in the revised manuscript. Below are our responses to the reviewers' comments. All reviewers' comments are in black, while the authors' responses are in blue. And all revisions in the revised manuscript are highlighted in red color. The influence of blocking and wave-train cold surges (CSs) on haze dispersion over eastern China is investigated in this study. The blocking CSs have relative weaker ability to remove the haze compared with the wave-train CSs. The topic aligns well with the scope of ACP. The manuscript is well-written with minor corrections of some comments.

1. The observed atmospheric visibility and relative humidity dataset are used to defined the occurrence of haze days in this study, with the threshold of 10 km visibility. However, the visibility observation in China was switched from manual observation to high temporal resolution automated observation since the year of 2013-2014. There are some systematic biases between manual and automated observation. And 7.5 Km automated observed visibility is suggested as the occurrence of haze. I would suggest re-defined the case of haze using different thresholds before and after the automated observation.

**Response:** Thanks for your suggestion. We use this definition to reproduce the evolution of haze during the occurrence of two types of cold surges, and its distribution is consistent with the results of our previous studies. It should be added that the cold surges and haze events involved from 2013 to 2016 have hardly changed after revising the definition of haze. Therefore, we modified the relevant contents and figures of haze definition. See lines 99-105: After filtering the other weather parameters affecting visibility (i.e., dust, precipitation, sandstorm), we defined a haze day as a day with visibility lower than 10 km and the Rhum less than 90 % occurring at any of the four times (02:00, 08:00, 14:00, and 20:00LT) (Yin et al., 2019a) from 1980-2013. However, the visibility observation in China was switched from manual observation to high temporal resolution automated observation after 2013 (Yin et al., 2017). Therefore, because of systematic biases between manual and automated observation, the 7.5 km automated observed visibility (Zhang et al., 2021) and Rhum less than 90 % are suggested as the occurrence of haze.

**References:**

- Yin, Z.C., Li, Y. Y., Wang, H. J.: Response of early winter haze in the North China Plain to autumn Beaufort sea ice, Atmos. Chem. Phys., 19, 1439–1453, https://doi.org/10.5194/acp-19-1439-2019, 2019a.
- Yin, Z. C., Wang, H. J.: Role of atmospheric circulations in haze pollution in December 2016., 17, 11673-11681, https://doi.org/10.5194/acp-17-11673-2017, 2017.
- Zhang X. Y., Yin, Z. C., Wang, H. J., and Duan M. K.: Monthly Variations of Atmospheric Circulations Associated with Haze Pollution in the Yangtze River Delta and North China, Adv. Atmos. Sci., 38(4), 569–580, https://doi.org/10.1007/s00376-020-0227-z, 2021.

**Figure S1.** (a) Spatial distribution of the annual haze days (day) in China averaged from 1980 to 2017. (b) Monthly variation of the regional-averaged haze days in the area of 22°N-37°N, 106°E-121°E.